# THINKING AUGMENTED PRE-TRAINING

## ABSTRACT

This paper introduces a simple and scalable approach to improve the data efficiency of large language model (LLM) training by augmenting existing text data with thinking trajectories. The compute for pre-training LLMs has been growing at an unprecedented rate, while the availability of high-quality data remains limited. Consequently, maximizing the utility of available data constitutes a significant research challenge. A primary impediment is that certain high-quality tokens are difficult to learn given a fixed model capacity, as the underlying rationale for a single token can be exceptionally complex and deep. To address this issue, we propose *Thinking augmented Pre-Training* (TPT), a universal methodology that augments text with automatically generated thinking trajectories. Such augmentation effectively increases the volume of the training data and makes high-quality tokens more learnable through step-by-step reasoning and decomposition. We apply TPT across diverse training configurations up to 100B tokens, encompassing pre-training with both constrained and abundant data, as well as mid-training from strong open-source checkpoints. Experimental results indicate that our method substantially improves the performance of LLMs across various model sizes and families. Notably, TPT enhances the data efficiency of LLM pre-training by a factor of 3. For a 3B parameter model, it improves the post-training performance by over $10\%$ on several challenging reasoning benchmarks.

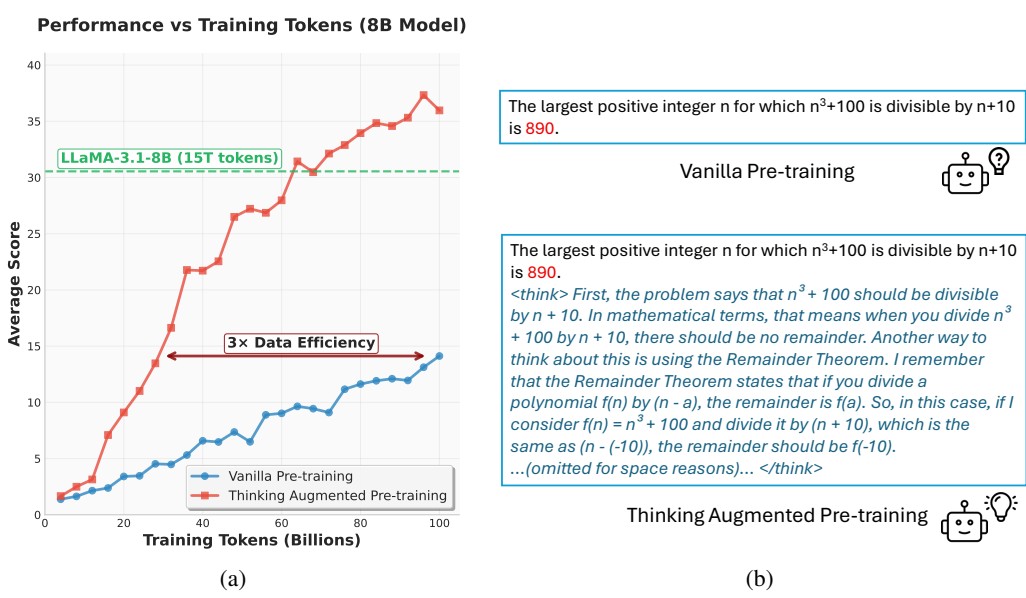

(a)                                                              (b)

Figure 1: (a) The average few-shot accuracy scores on the GSM8k and MATH datasets with respect to total training tokens. Both models are pre-trained from scratch with $8$B parameters. One model employed a vanilla next-token prediction objective, while the other utilized thinking-augmented pre-training. (b) Illustration of a thinking augmented data sample. The token in red, "890", is both correct and valuable, yet it is difficult to learn directly. The complete text is provided in Appendix Table 8.

# 1 INTRODUCTION

Large language models (LLMs) have achieved remarkable success across a wide range of tasks, including natural language understanding, code generation, and complex reasoning (Brown et al., 2020; Hurst et al., 2024; Dubey et al., 2024). A foundational principle underpinning this success is the scaling law (Kaplan et al., 2020), which posits that increasing the model size and the amount of training tokens leads to continuous performance improvements. Nevertheless, LLMs are data hungry, with recently released open-source models being trained on over 10 trillion tokens (Dubey et al., 2024; Yang et al., 2025) primarily derived from web-crawled corpora. The development of the next generation of LLMs necessitates scaling up both compute and training data. Although the compute has kept growing and is projected to continue this trajectory in the foreseeable future, the pool of human-authored, organically generated data on the web is finite and has been largely exhausted by existing frontier models. Consequently, as the scale of LLMs expands, the challenge of curating and leveraging high-quality data intensifies, making data engineering a central aspect of model development.

Modern data engineering pipelines (Penedo et al., 2024; AI et al., 2025; Li et al., 2024) for large-scale pre-training are multifaceted processes, involving the acquisition of raw data from diverse sources, including web crawls, code repositories, and books, among others. These pipelines often employ techniques such as parsing, deduplication, filtering, domain balancing, rewriting (Maini et al., 2024), and synthetic data generation (Gunasekar et al., 2023) to enrich the quality and diversity of the resulting training corpus.

Orthogonal to the development of enhanced data curation pipelines, a critical but underexplored dimension is the maximization of utility from existing data. Prior research addresses this challenge through a data selection lens (Lin et al., 2024; Mindermann et al., 2022), proposing to train models exclusively on a subset of valuable tokens that are learnable but are not yet learned. However, some valuable tokens can be exceptionally difficult to learn in a single next-token prediction step, as they often represent the outputs of intricate, multi-step human reasoning processes (Xiang et al., 2025). Figure 1 provides an illustrative example where the correct answer token "890" is derived from a sequence of reasoning steps that necessitate an understanding of polynomial division, the Remainder Theorem, and the properties of divisors. When a model's capacity is limited, it may struggle to learn such tokens beyond pure memorization, which will not generalize well.

To circumvent these limitations, we introduce a thinking augmented training approach called TPT that automatically expands pre-training datasets and enhances their learnability for LLMs. Our method augments the raw data by generating thinking trajectories using readily available open-source LLMs. These trajectories simulate an expert's in-depth thought process as they analyze the given text, mirroring the way humans learn new knowledge. Given that explanation is often easier than generation from scratch, models trained on such augmented data can, as our experiments demonstrate, surpass the performance of the thinking generation model itself. TPT is highly scalable as it requires no human annotation and imposes no constraints on document structure.

Thinking pattern analysis reveals that our method naturally up-samples high-quality data, aligning with contemporary data engineering practices that have been empirically validated as effective. For example, thinking trajectories tend to be longer in domains such as mathematics. A positive correlation exists between the reasoning intensity and difficulty of the original text and the thinking length. A longer thinking length implies more training compute allocated to the corresponding tokens. This bears a resemblance to test-time scaling (Jaech et al., 2024) where more difficult samples benefit from increased inference compute. The key distinction is that we apply this principle during training, allocating more training compute to challenging samples, which in turn enhances their learnability for models. RPT (Dong et al., 2025) operates in a similar spirit by applying reinforcement learning to next-token prediction during pre-training, but it necessitates substantially more compute due to online rollouts and a token-level training paradigm.

We evaluate our approach across various settings with up to 100B training tokens. These settings include pre-training from scratch under both constrained and abundant data regimes, as well as mid-training from extensively pre-trained checkpoints. Our experiments demonstrate significant improvements in both the data efficiency and the final performance of LLMs across different model sizes and evaluation benchmarks, spanning math, code, and general reasoning. To achieve the same level of base model performance, TPT reduces the required training tokens by a factor of 3,

underscoring its effectiveness in maximizing the utility of existing data. For supervised fine-tuning on a public dataset with 350k samples, thinking augmented mid-training provides consistent performance boosts, particularly for models not heavily pre-trained on reasoning-intensive data.

Our main contributions can be summarized as follows:

- We propose thinking augmented pre-training, a simple and scalable data engineering approach that transforms pre-training datasets into a format conducive to LLM learning.
- We provide an empirical analysis of the generated thinking trajectories, showing that TPT naturally up-samples high-quality data and dynamically allocates training compute based on the domain, difficulty, and reasoning intensity of the raw text.
- Extensive experiments demonstrate that our proposed method significantly enhances data efficiency and model performance across a range of training setups, including pre-training from scratch, mid-training, and supervised fine-tuning.

## 2 THINKING AUGMENTED PRE-TRAINING

**Prompt: Thinking Trajectory Generation**

```
{{CONTEXT}}
## End of the context

Simulate an expert's in-depth thought process as they analyze the above
context, focusing on complex and informative aspects.  Skip trivial
details.  Use Feynman technique whenever possible to ensure a deep
understanding.
```

Given a document $d$ from the pre-training dataset, a thinking trajectory $t$ is generated using an off-the-shelf model with the specified prompt, where the placeholder `{{CONTEXT}}` is replaced by the document text. The original document and the generated thinking trajectory are concatenated to form the augmented training sample $x = [d; t]$. We then minimize the standard next-token prediction loss to train LLMs on this augmented dataset.

$$\min \mathcal{L} = -\frac{1}{N} \sum_{i=1}^{N} \log p(x_i \mid x_{<i}) \tag{1}$$

where $N$ is the total number of tokens in the sample $x$. This approach is applicable across different LM training stages, including pre-training from scratch and mid-training from existing checkpoints.

Our proposed method offers several compelling properties:

- **Scalability** The process of thinking augmentation is extremely simple and universally applicable to any text data. Compared to RPT (Dong et al., 2025), our method does not require online rollouts and operates at the document level, which makes it highly scalable.
- **Dynamic Allocation of Training Compute** Valuable tokens can be difficult to learn in a generalizable manner by training on them directly, as exemplified in Figure 1. Thinking augmentation breaks down complex tokens into smaller, more explainable steps, thereby effectively allocating more training compute to them. This is analogous to test-time scaling but applied during training instead of inference. Empirical evidences in Section 4 shows that thinking trajectories tend to be longer for high-value domains and documents, which functions as a natural up-sampling mechanism.
- **LLM-friendly Data Format** Web-crawled data are often noisy and of varying quality, necessitating extensive filtering and rewriting (Li et al., 2024; Maini et al., 2024). TPT provides a complementary method to transform raw text into a more LLM-friendly format that facilitates more efficient learning.

## 3 EXPERIMENTS

To comprehensively evaluate the efficacy of our proposed method, we conduct experiments across various training configurations, including pre-training from scratch under both constrained and

abundant data regimes, as well as mid-training from existing checkpoints. For all experiments, our training corpora consist of MegaMath-Web-Pro-Max (Zhou et al., 2025; Wang et al., 2025) and FineWeb-Edu (Penedo et al., 2024). Documents were packed into samples of 8k tokens each, and sample weights were adjusted to balance the different data sources. Further details and training hyperparameters are provided in Appendix A.1.

### 3.1 PRE-TRAINING UNDER ABUNDANT DATA

For pre-training under abundant data, each data sample is utilized at most once, assuming the dataset has been deduplicated. This setting aligns with many current LLM pre-training configurations, where the compute is the main bottleneck.

In this section, we train two 8B parameter models from scratch, following the LLaMA-3-8B architecture, with a total training budget of 100B tokens. The only distinction is that one is trained on the original dataset, whereas the other is trained on the thinking augmented dataset. Both models are trained for 25k steps using a batch size of 4M tokens. The training epochs are both less than 1 due to the large scale of the dataset. In terms of raw documents processed by the models, the vanilla model is exposed to approximately $3\times$ documents than the thinking augmented model.

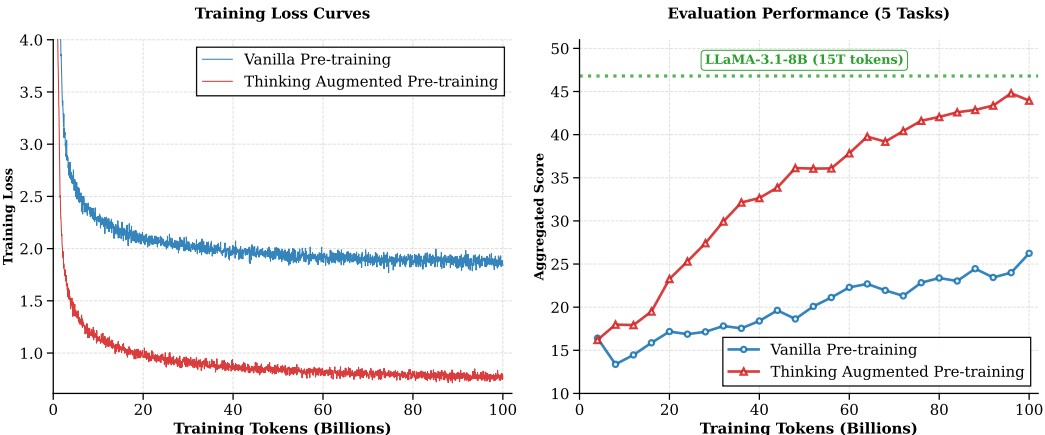

Figure 2: Pre-training loss curves and aggregated scores on 5 tasks with respect to total training tokens (8B model). Both models are trained from scratch on 100B tokens. The loss values are not directly comparable due to differences in data distributions, but we demonstrate how thinking augmentation reduces data noise and enhances learnability. The final scores of both models are detailed in Appendix Table 5.

In Figure 2, the training loss of the thinking augmented model is substantially lower than that of the vanilla model, suggesting that the augmented data is less noisy and more readily learnable for LLMs. However, the lower perplexity alone does not guarantee superior performance on downstream tasks. Therefore, we also monitor the aggregated score across 5 tasks: GSM8k (5-shot with CoT), MATH (4-shot with CoT), BoolQ (0-shot), MMLU (2-shot with CoT), and MMLU$_{Pro}$ (2-shot with CoT) (Cobbe et al., 2021; Hendrycks et al., 2021b;a; Wang et al., 2024; Clark et al., 2019). During evaluation, failure of answer extraction will result in a zero score. Consequently, multiple-choice tasks may see a lower score than random guessing at the early training stage.

The thinking augmented pre-training initially performs similarly to the vanilla model but rapidly surpasses it after 20B tokens, and this performance gap continues to widen. At 100B training tokens, the thinking-augmented model achieves a score comparable to that of LLaMA-3.1-8B, which was trained on $150\times$ more data (15T tokens).

In Table 1, we assess the models on a suite of more challenging reasoning benchmarks after SFT. The results reveal that vanilla pre-training fails to develop strong reasoning capabilities, as evidenced by the model's very low scores on benchmarks like AIME24 and LiveCodeBench (LCB). In stark contrast, TPT provides a substantial performance uplift across all evaluated tasks, even outperforming LLaMA-3.1-8B-Instruct on every benchmark. This outcome highlights a promising improvement in

Table 1: Performance after supervised fine-tuning on the 2B-token Mixture-of-Thoughts dataset.

| Model | # tokens | AIME24 | MATH-500 | LCB | GPQA | MMLU_pro |
|---|---|---|---|---|---|---|
| Vanilla-8B→SFT | 100B + 2B | 1.0 | 33.8 | 1.9 | 27.7 | 29.0 |
| TPT-8B→SFT | 100B + 2B | **35.2** | **82.4** | **23.4** | **45.2** | **59.8** |
| LLaMA-3.1-8B-Instruct | 15T | 5.4 | 49.4 | 9.4 | 31.4 | 43.6 |

data efficiency, demonstrating that TPT enables models to achieve superior reasoning abilities with a fraction of the conventional training data.

## 3.2 PRE-TRAINING UNDER CONSTRAINED DATA

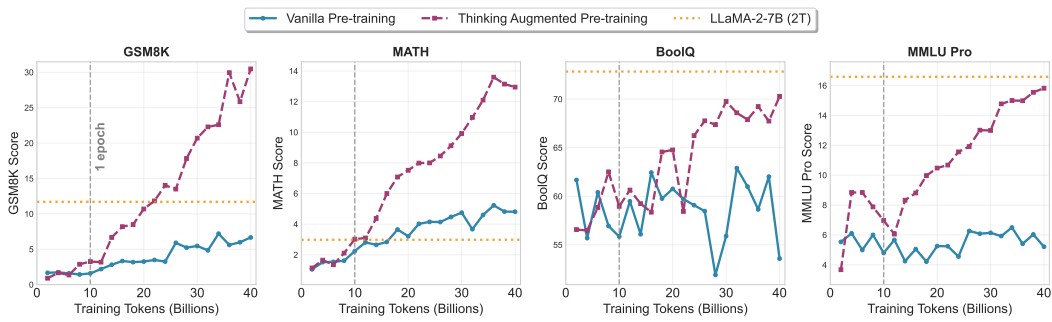

Figure 3: Task scores with respect to total training tokens (8B model). The tokens in raw documents are constrained to 10B via random sampling. The final scores are detailed in Appendix Table 7.

Frontier LLM training is approaching the exhaustion of high-quality web data, making LM scaling under constrained data a critical concern (Muennighoff et al., 2023). To simulate this scenario, we limit the total number of training tokens from raw documents to 10B and set the training budget to 40B tokens. We train two 8B parameter models from scratch, one using vanilla LM pre-training and the other with our proposed thinking augmented pre-training. Consequently, the vanilla model sees the entire dataset 4 epochs, while the thinking augmented model sees the data only once due to the increased token count after augmentation.

Figure 3 illustrates the performance of both models. Initially, they exhibit similar performance trajectories across all benchmarks. However, as training progresses, a clear divergence emerges: the performance of vanilla pre-training model plateaus or improves slowly as unique tokens are exhausted, whereas TPT continues to improve steadily. This divergence is particularly notable on mathematical reasoning tasks such as GSM8k and MATH. The sustained improvement suggests that thinking trajectories enable models to extract more value from the same underlying data.

## 3.3 THINKING AUGMENTED MID-TRAINING

Mid-training, alternatively referred to as continual pre-training, enhances the capabilities of existing LLMs by further training on carefully curated datasets. This methodology circumvents the need to train models from scratch, making it a cost-effective strategy.

We apply thinking augmented mid-training to three open-source models, ranging from 1.5B to 7B parameters, encompassing two model families: Qwen2.5 (Yang et al., 2024) and LLaMA-3 (Dubey et al., 2024). All models are initialized from their respective base versions. Initially, each model was trained on 100B tokens of thinking augmented data, followed by supervised fine-tuning (SFT) on the publicly available Mixture-of-Thoughts dataset (HuggingFace, 2025) to align the models with a chat format. This SFT dataset comprises 350k samples distilled from DeepSeek-R1 (Guo et al., 2025). The specific hyperparameters for both mid-training and SFT are documented in Appendix Table 4.

Table 2: Mid-training results on math, code, and general reasoning benchmarks after supervised fine-tuning. Models denoted with an asterisk (*) were trained by us using the public Mixture-of-Thoughts dataset. [†]: DS-Distill-Qwen-7B, the model used for generating thinking trajectories, is not directly comparable as it was trained on a larger SFT dataset.

| Model | Math | | | | | Code | | General | | |
| --- | --- | --- | --- | --- | --- | --- | --- | --- | --- | --- |
| | MATH-500 | AIME24 | AIME25 | GSM8k | HMMT | HEval | LCB v4_v5 | GPQA-D | MMLU$_{Pro}$ | JEEBench |
| *Qwen2.5-1.5B based Models* | | | | | | | | | | |
| Qwen2.5-1.5B-Instruct | 52.6 | 3.1 | 0.7 | 71.0 | 0.0 | 59.8 | 4.5 | 26.5 | 28.7 | 15.6 |
| OpenR1-Qwen2.5-1.5B* | 79.6 | 20.8 | 22.3 | 79.2 | 10.3 | 45.1 | 11.6 | 36.8 | 43.5 | 38.2 |
| TPT-Qwen2.5-1.5B | **82.3** | **28.5** | **25.6** | **80.1** | **14.0** | **63.4** | **17.2** | **40.0** | **50.5** | **50.3** |
| *LLaMA-3.2-3B based Models* | | | | | | | | | | |
| LLaMA-3.2-3B-Instruct | 40.6 | 3.8 | 0.3 | 73.6 | 0.3 | 57.9 | 2.0 | 28.9 | 32.4 | 14.0 |
| OpenR1-LLaMA-3B* | 59.8 | 5.8 | 7.1 | 69.1 | 6.3 | 45.7 | 13.9 | 32.8 | 45.8 | 26.6 |
| TPT-LLaMA-3B | **75.5** | **18.6** | **17.5** | **81.6** | **11.7** | **65.2** | **20.0** | **41.7** | **55.5** | **42.4** |
| *Qwen2.5-7B based Models* | | | | | | | | | | |
| Qwen2.5-7B-Instruct | 74.1 | 8.1 | 5.1 | 89.6 | 2.3 | 84.1 | 13.8 | 32.1 | 54.6 | 32.7 |
| OpenR1-Qwen2.5-7B | 89.0 | 50.5 | 34.8 | 87.6 | **28.0** | 81.7 | **39.4** | 52.1 | 62.8 | 69.1 |
| TPT-Qwen2.5-7B | **92.5** | **57.5** | **39.4** | **90.8** | 25.0 | **89.0** | 39.3 | **54.7** | **64.0** | **73.6** |
| *For Reference Only* | | | | | | | | | | |
| DS-Distill-Qwen-7B[†] | 93.5 | 53.2 | 35.5 | 90.7 | 21.7 | 87.8 | 36.2 | 49.0 | 56.4 | 49.9 |
| GPT-4o | 74.6 | 9.3 | 14.0 | 95.0 | 5.8 | 90.2 | 32.9 | 49.9 | 72.6 | 44.8 |

We evaluate the models on a diverse set of 10 challenging benchmarks, covering mathematical reasoning (MATH-500, AIME24, AIME25, GSM8k, HMMT), code generation (HumanEval and LiveCodeBench v4_v5), and general knowledge reasoning (GPQA-Diamond, MMLU$_{Pro}$, JEEBench). For a fair comparison, we primarily compare our models against the OpenR1 models, which are fine-tuned on the same Mixture-of-Thoughts dataset. During inference, we set the maximum generation length to 32k tokens to allow enough space for thinking.

The results after supervised fine-tuning are summarized in Table 2. TPT models substantially outperform their counterparts. This improvement is particularly pronounced for the LLaMA-3 series. For instance, the 3B LLaMA model demonstrates remarkable improvements on mathematical benchmarks like AIME24 (from $5.8\%$ to $18.6\%$, a $3\times$ increase) and general reasoning tasks. The larger performance boost observed in LLaMA models likely originates from their pre-training corpora containing less reasoning-intensive data compared to Qwen2.5. The consistent improvements across all model sizes, from 1.5B to 7B parameters, underscore the scalability and robustness of TPT.

## 4 ANALYSIS OF THINKING PATTERNS

To gain more insights into the generated thinking trajectories, we utilize the metadata provided by the *essential-web-v1.0* dataset (AI et al., 2025) to examine the influence of thinking augmentation on the training data distribution. A balanced sample of 20k documents was drawn, stratified across three metadata groups: domain, reasoning intensity, and target audience. Subsequently, thinking trajectories for these documents were generated using DeepSeek-R1-Distill-Qwen-7B, adhering to the same prompting methodology as outlined in Section 2.

Figure 4 presents the length distribution grouped by the domain tag. Domains such as *Mathematics* and *Physics* exhibit notably longer thinking trajectories, which aligns with the a priori expectation that these fields necessitate deep reasoning. Furthermore, a clear positive correlation is observed between reasoning intensity and thinking length, with the "Advanced Reasoning" group possessing approximately $50\%$ more tokens than the "No Reasoning" group. Somewhat counterintuitively, for the target audience tag, the "Expert" group exhibits shorter thinking trajectories compared to the "Undergraduate" group. This may be attributed to the fact that expert-level documents often contain

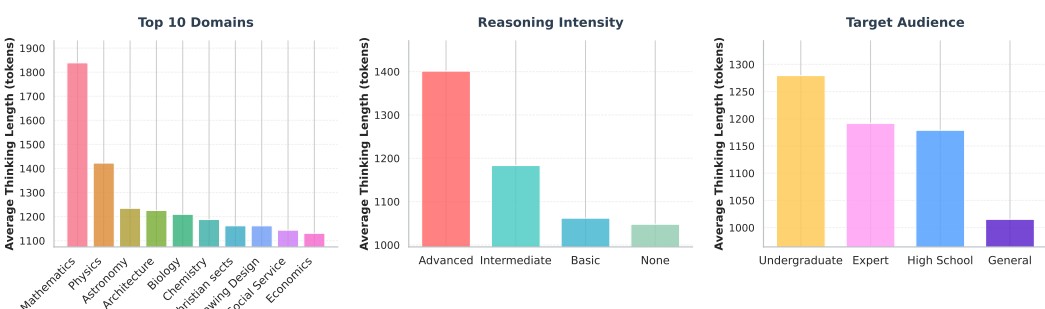

Figure 4: The average number of thinking tokens, categorized by domain, target audience, and reasoning intensity. The figure lists only the top-10 domains that exhibit the longest thinking trajectories.

more specialized concepts, but do not necessarily require a greater number of reasoning steps for comprehension.

Our empirical analysis demonstrates that high-value data naturally yields longer thinking trajectories. This results in more training compute being allocated to them, thereby effectively up-sampling valuable content without the use of manual heuristics or explicit quality filtering mechanisms. Similar observations are also made by Guha et al. (2025), where they find filtering based on GPT-4.1 response length is the most effective way among several baselines for improving the math question selection for post-training. Several illustrative examples are provided in Appendix A.6.

## 5 ABLATION STUDIES

**Thinking Trajectory Generation** We explore several alternative strategies for generating thinking trajectories, comparing them against our default methodology.

- **Customized Back-thinking Model:** Using an SFT dataset, we fine-tune DeepSeek-R1-Distill-Qwen-7B to generate thinking content within the `<think>` and `</think>` tags with the final response and the original question serving as input. During data generation, documents from the pre-training dataset are provided as input to this back-thinking model.

- **Prompt with Random Focus Point:** We modify the prompt in Section 2 by instructing the model to focus on a random point within the document. The purpose is to see if random focus can help the model generate more diverse outputs. The complete prompt is included in Appendix A.3.

Table 3: Ablation for thinking data generation. All models first undergo 40B tokens of thinking augmented mid-training, followed by SFT on the Mixture-of-Thoughts dataset.

|  | Math | | | Code | | General | | |
|---|---|---|---|---|---|---|---|---|
|  | MATH-500 | AIME24 | AIME25 | HEval | LCB | GPQA | MMLU$_{\text{Pro}}$ | JEE |
| TPT-LLaMA-3B (40B tokens) | 72.0 | 11.7 | 15.0 | 64.6 | 18.7 | 37.7 | 52.3 | 36.0 |
| w/ back-thinking model | 73.8 | 14.7 | 13.8 | 57.3 | 18.4 | **38.4** | 53.9 | **41.7** |
| w/ random focus point | 72.2 | 12.4 | 14.4 | **67.1** | 18.9 | 38.3 | 53.5 | 36.7 |
| *Using smaller model for thinking generation* | | | | | | | | |
| w/ DS-Distill-Qwen-1.5B | **75.5** | **17.7** | **16.5** | 64.6 | **21.4** | 36.6 | **54.1** | 41.3 |

The results in Table 3 indicate that, while alternative thinking generation strategies yield slight improvements over our default method, both the custom back-thinking model and the random focus point strategy demonstrate only marginal gains across most benchmarks. Additionally, these approaches introduce extra implementation complexity due to the need for custom fine-tuning or dynamic prompt modification. Consequently, we stick to the default strategy for our main experiments to ensure simplicity and reproducibility.

**Scaling Thinking Generation Model** In Table 3, perhaps surprisingly, the results reveal that using a smaller model (DeepSeek-R1-Distill-Qwen-1.5B) for thinking generation outperforms the default

7B model. This finding aligns with observations reported by OpenThoughts (Guha et al., 2025). The smaller model may generate trajectories that are better suited for downstream model learning. The relationship between the model being fine-tuned and the model used for thinking generation warrants further investigation.

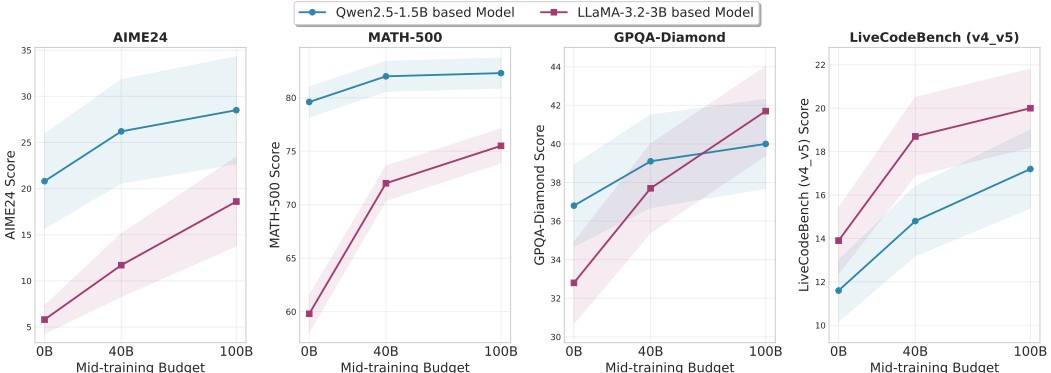

Figure 5: Task scores with respect to the mid-training token budget. The "0B" data point corresponds to direct SFT without thinking augmented mid-training.

**Impact of Mid-training Token Budget** As depicted in Figure 5, SFT with 350k samples proves insufficient for developing strong reasoning capabilities. For instance, the LLaMA-3B model without mid-training barely solves any AIME24 problems, whereas the same model demonstrates a substantial $\sim$ 15-point performance increase following 100B tokens of thinking augmented mid-training. This observation underscores the critical role of mid-training in the cultivation of reasoning abilities that are otherwise difficult to achieve through SFT alone.

As we scale the mid-training token budget from 0 to 100B tokens, there are continual performance gains across both model sizes and all evaluated tasks. The sustained upward trend suggests that scaling beyond 100B tokens would likely yield further improvements, indicating that our approach benefits from increased training compute.

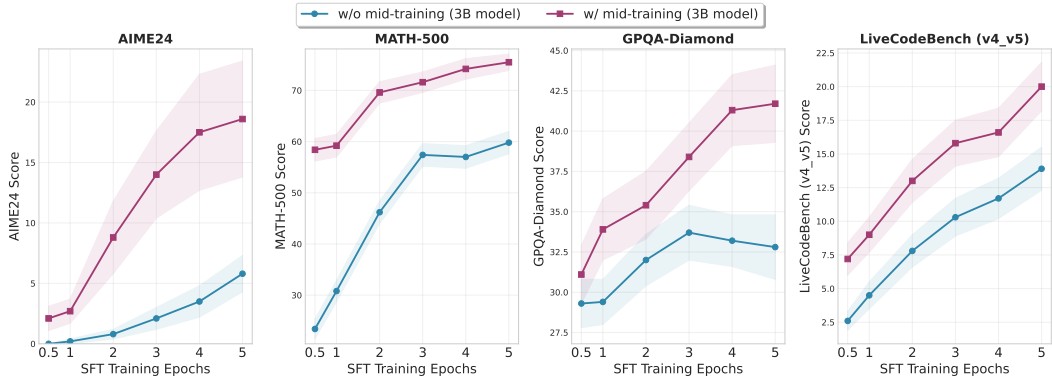

Figure 6: Task scores with respect to SFT epochs. The "w/o mid-training" variant is initialized from the LLaMA-3.2-3B-Base checkpoint.

**Impact of SFT Data Size** Figure 6 shows that increasing SFT epochs generally improves performance across most benchmarks, with no serious overfitting observed even at 5 epochs. Performance on LiveCodeBench appears not to have fully converged, suggesting the potential for further gains with extended training. We maintain 5 epochs for our study to align with established practices.

More importantly, checkpoints that underwent thinking augmented mid-training demonstrate superior starting points compared to their vanilla counterparts. This advantage persists throughout the SFT

phase, highlighting how thinking augmentation creates a stronger foundation that amplifies the benefits of subsequent fine-tuning phases.

## 6 RELATED WORK

**Data Engineering for Large-Scale Pre-training** A core contributing factor to the success of large foundation models is the curation of large-scale, high-quality training data. The scaling laws (Kaplan et al., 2020) suggest that model performance can be significantly improved by increasing the size of the training dataset, alongside the model size. The total training tokens for LLMs have been scaled substantially, from 300B tokens for GPT-3 (Brown et al., 2020) to over 10T tokens (Dubey et al., 2024; Yang et al., 2025) in just a few years. The modern data curation pipeline is a complex, multi-stage process designed to transform raw data into a high-quality corpus. For example, the FineWeb dataset (Penedo et al., 2024) first extracts text content from Common Crawl, then performs deduplication, and subsequently applies a series of heuristic and model-based filters to remove low-quality and harmful content. DCLM (Li et al., 2024) introduces a modular evaluation framework to provide a unified testbed for pre-training data curation.

As the pre-training of foundation models continues to scale, the community is moving towards exhausting high-quality human-authored data on the web. As such, synthetic data generation has emerged as a promising approach for both pre-training (Gunasekar et al., 2023; Lin et al., 2025; Qin et al., 2025) and post-training (Guha et al., 2025; Liu et al., 2025). For example, the Phi (Gunasekar et al., 2023) series of models heavily rely on textbook-like synthetic data generated by GPT-3.5 and GPT-4. Prior studies (Maini et al., 2024; Nguyen et al., 2025; Allen-Zhu & Li, 2024) have also shown that rewriting raw text data, through techniques such as paraphrasing, can enhance data quality and thereby improve model performance. Our proposed method augments existing datasets by generating detailed thinking trajectories, and is orthogonal to rewriting based approaches. Reasoning CPT (Ishibashi et al., 2025) closely relates to our work by carefully prompting a non-thinking LLM to mine hidden thoughts, and show improvements on the MMLU benchmark (Hendrycks et al., 2021a). However, Reasoning CPT only scales to $\sim 150M$ training tokens under a continual training setup, and the evaluation is limited to base models without post-training. BoLT (Ruan et al., 2025) focuses on bootstrapping latent thoughts using EM algorithm. In contrast, our method scales to 100B tokens for both pre-training and mid-training of LLMs, and demonstrates significant improvements on a wide range of challenging benchmarks.

**Chain-of-Thought (CoT) Reasoning** enables LLMs to generate intermediate steps for solving complex problems, thereby eliciting their reasoning capabilities (Wei et al., 2022) at the cost of increased inference time. Initial studies demonstrated that simply encouraging a step-by-step process dramatically improves performance on reasoning tasks. Subsequent research quickly moved beyond linear chains, exploring more sophisticated structures like trees (Yao et al., 2023) to allow for exploration, backtracking, and self-correction.

Instead of solely relying on prompting techniques, OpenAI o1 (Jaech et al., 2024) and DeepSeek-R1 (Guo et al., 2025) propose to fine-tune LLMs with reinforcement learning to explicitly encourage the generation of long thinking trajectories. These methods demonstrate substantial performance improvements on solving Olympiad-level math and coding problems (Chen et al., 2021), and observe a positive correlation between the length of the generated tokens and task performance, a phenomenon usually referred to as test-time scaling (Jaech et al., 2024). In this work, we leverage open-source LLMs to generate thinking trajectories for augmenting training data. We show that training with such data significantly improves the reasoning capabilities of LLMs across various training stages.

## 7 CONCLUSION

In this work, we introduce *Thinking augmented **P**re-**T**raining* (TPT), a simple and scalable approach to enhance the pre-training data efficiency by augmenting existing text data with thinking trajectories. Experimental results with up to 100B training tokens demonstrate that TPT substantially improves both data efficiency and final model performance. The method yields consistent gains across different model sizes and training configurations, with particularly notable improvements in reasoning-intensive tasks. We hope our findings will inspire continued research into scalable data engineering that maximize the potential of foundation models while making more efficient use of data.

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

# A APPENDIX

## A.1 TRAINING DETAILS

**Pre-training and Mid-training Datasets** The pre-training and mid-training phases utilize a composite of two datasets: FineWeb-Edu (Penedo et al., 2024) and MegaMath-Web-Pro-Max (Zhou et al., 2025; Wang et al., 2025). FineWeb-Edu is a high-quality subset of the FineWeb dataset that focuses on educational content. We de-duplicate the dataset with exact matching at the document level. MegaMath-Web-Pro-Max is a filtered subset of the MegaMath dataset, comprising approximately 70B tokens of math-intensive content from the web. When mixing the two datasets, we apply a sample weight of 0.125 to FineWeb-Edu to balance the domain distribution.

**Post-training Datasets** For supervised fine-tuning (SFT), the public Mixture-of-Thoughts dataset (HuggingFace, 2025) is employed, containing 350k examples covering math, coding, and science domains. This dataset has been reported to replicate the performance of *DeepSeek-R1-Distill-Qwen-7B* (Guo et al., 2025), which is trained on a private dataset of 800k examples. However, our reproduction experiments show that, while the performance of 7B models is comparable, the 1.5B model remains inferior to *DeepSeek-R1-Distill-Qwen-1.5B*, which can likely be attributed to the smaller dataset size.

**Thinking Trajectory Generation** The input document length is truncated to a maximum of 2k tokens, with the maximum number of generated thinking tokens set to 8k. A temperature of 0.6 and a top-p of 0.9 are employed for generation. We do not generate beyond the end of thinking tag (e.g., `</think>`) as the final response is often the summary of the thinking process without introducing new information.

For mid-training experiments, we utilize *DeepSeek-R1-Distill-Qwen-7B* as the thinking generation model. Despite not being the strongest model available, it has an open-source post-training recipe provided by OpenR1 (HuggingFace, 2025), which facilitates fair comparisons with other baselines. For pre-training from scratch, we use *Qwen3-8B* (Yang et al., 2025) to generate thinking trajectories. The entire data generation pipeline takes roughly 20k A100 GPU hours to yield enough data for 100B training tokens.

Table 4: Hyperparameters for pre-training, mid-training, and SFT post-training.

| | Pre-training | Mid-training | | | SFT Post-training | | |
|---|---|---|---|---|---|---|---|
| | 8B | 1.5B | 3B | 7B | 1.5B | 3B | 7B |
| Batch size (# tokens) | 4M | 8M | 8M | 8M | 1M | 1M | 1M |
| Learning rate | 3e-4 | 6e-5 | 4e-5 | 3e-5 | 6e-5 | 4e-5 | 3e-5 |
| LR Schedule | Constant | Cosine | Cosine | Cosine | Cosine | Cosine | Cosine |
| # GPUs | 32 | 16 | 32 | 32 | 8 | 8 | 8 |
| Max sequence length | 8k | 8k | 8k | 8k | 32k | 32k | 32k |
| Max steps | 25k | 12.5k | 12.5k | 12.5k | - | - | - |
| # of epochs | - | - | - | - | 5 | 5 | 5 |
| Gradient clipping | 1.0 | 0.3 | 0.3 | 0.3 | 0.2 | 0.2 | 0.2 |
| Weight decay | 0.1 | 0.1 | 0.1 | 0.1 | 0 | 0 | 0 |
| Warmup step | 100 | 100 | 100 | 100 | 400 | 400 | 400 |
| Adam $\beta_1$ | 0.9 | 0.9 | 0.9 | 0.9 | 0.9 | 0.9 | 0.9 |
| Adam $\beta_2$ | 0.95 | 0.95 | 0.95 | 0.95 | 0.999 | 0.999 | 0.999 |

**Training Hyperparameters** The hyperparameters for pre-training, mid-training, and SFT post-training are summarized in Table 4. All the training jobs are performed on MI300 GPUs. It takes about 1 week to pre-train an 8B model for 100B tokens. Similarly, mid-training for the 7B model also takes about 1 week. For SFT post-training with the Mixture-of-Thoughts dataset, fine-tuning a 7B model for 5 epochs takes about 3.5 days.

Since the SFT dataset contains examples with significantly varying lengths, data packing is employed to improve training efficiency. Example boundaries are respected to avoid cross-example attention contamination. The loss is only computed over the assistant response tokens.

**Model Initialization** For pre-training, the model weights are randomly initialized following the LLaMA-3 model architecture. For mid-training, the {1.5B, 3B, 7B} models are initialized from {Qwen2.5-Math-1.5B, Llama-3.2-3B, Qwen2.5-Math-7B}, respectively. To support an extended context length from 8k to 32k during post-training, the RoPE base frequency is multiplied by 16.

## A.2 EVALUATION DETAILS

**Base Model Evaluation** For evaluating the base models, we report the average performance across 5 datasets: MMLU (Hendrycks et al., 2021a), MMLU$_{Pro}$ (Wang et al., 2024), BoolQ (Clark et al., 2019), GSM8K (Cobbe et al., 2021), and MATH (Hendrycks et al., 2021b).

We use 5-shot and 4-shot prompts for GSM8K and MATH, respectively, while utilizing 2-shot CoT prompts for MMLU and MMLU$_{Pro}$, and a zero-shot prompt for BoolQ. We use regular expressions to extract the final answer from the model output, and treat the answer as incorrect if no valid answer is found. As a result, some reported scores are lower than random guessing for multiple-choice datasets due to the model failing to produce a valid option. This phenomenon is particularly pronounced in models trained with less tokens. The maximum number of tokens generated is capped at 2k. Answer accuracy is the metric reported for all datasets.

**Instruction-tuned Model Evaluation** For models that have undergone post-training, we evaluate their performance on a set of more challenging benchmarks, which includes MATH-500 (Lightman et al., 2024), AIME 2024 (AIME24) and AIME 2025 (AIME25), HMMT 2025 (HMMT) (Balunović et al., 2025), GPQA-Diamond (GPQA) (Rein et al., 2024), MMLU-Pro (MMLU$_{Pro}$) (Wang et al., 2024), HumanEval (HEval) (Chen et al., 2021), LiveCodeBench v4 and v5 (LCB) (Jain et al., 2024), and JEEBench (Arora et al., 2023).

We report *Pass@1* as the main metric. For coding tasks, a set of test cases is provided to automatically verify the correctness of the generated code. To reduce the variance of the *Pass@1* metric on smaller datasets, we generate multiple samples per question and subsequently compute the average *Pass@1* across all samples. Specifically, we generate 64 samples for AIME24 and AIME25, 16 samples for LiveCodeBench, 8 samples for GPQA-Diamond, and 4 samples for MATH-500. The maximum number of thinking tokens is set to 32k.

For the majority of our evaluations, we leverage the open-source *lighteval* [1] library. For a few datasets that are not yet supported, including HumanEval, HMMT, and JEEBench, we use the evaluation scripts provided by *evalchemy* [2]. For all generative tasks, we set the sampling temperature to 0.6 and the top-$p$ value to 0.95. The evaluation is conducted on 4 A100 GPUs and takes about 1 day for the 8B model.

## A.3 PROMPT TEMPLATES

> **Prompt: Thinking Trajectory Generation with Random Focus Point**
>
> ```
> {{RANDOM CONTEXT PREFIX}}
> «<READING HERE»>
> {{REMAINING CONTEXT}}
> ## End of the context
>
> An expert is focused at the «<READING HERE»> position.  Simulate the
> expert's in-depth thought process as they analyze the above context,
> focusing on complex and informative aspects.  Skip trivial details.
> Use Feynman technique whenever possible to ensure a deep understanding.
> ```

The prompt template with a random focus point is provided above. For each document, we randomly select a position to serve as the focus point, and subsequently partition the document into two distinct parts.

Table 5: Base model performance across 5 datasets and their average.

| Model | # tokens | GSM8k | MATH | BoolQ | MMLU | MMLU$_{pro}$ | Avg |
|-------|----------|-------|------|-------|------|--------------|-----|
| Vanilla-8B | 100B | 19.2 | 9.1 | 66.5 | 26.2 | 10.3 | 26.2 |
| TPT-8B | 100B | **50.1** | **21.8** | **75.0** | **46.7** | **26.2** | **43.9** |
| LLaMA-2-7B | 2T | 11.7 | 3.0 | 72.8 | 40.0 | 16.6 | 28.8 |
| LLaMA-3.1-8B | 15T | 47.0 | 14.1 | 83.5 | 57.6 | 31.8 | 46.8 |

## A.4 ADDITIONAL RESULTS

The final per-task results in Table 5 demonstrate the remarkable effectiveness of TPT-8B, particularly in the domain of mathematical reasoning. Despite being trained on only 100B tokens, TPT-8B achieves substantial improvements over the vanilla baseline. Specifically, performance on GSM8k increases from 19.2% to 50.1%, and MATH scores more than double, rising from 9.1% to 21.8%. These gains are particularly noteworthy when compared to LLaMA-3.1-8B, which necessitated a significantly larger volume of training data to attain comparable performance in mathematical reasoning.

Table 6: Impact of vanilla mid-training and scores of other open-source models. For "vanilla mid-training → SFT", we continually train the LLaMA-3.2-3B model on the text data for 40B tokens without thinking augmentation, and then perform SFT on the Mixture-of-Thoughts dataset. The distillation based models are trained on a larger private dataset with 800k samples and are therefore not directly comparable.

| | Math | | | Code | | General | | |
|---|---|---|---|---|---|---|---|---|
| | MATH-500 | AIME24 | AIME25 | HEval | LCB | GPQA | MMLU$_{Pro}$ | JEEBench |
| *Impact of Vanilla Mid-training (40B token budget)* | | | | | | | | |
| Vanilla mid-training → SFT | 59.8 | 5.0 | 6.3 | 36.0 | 5.7 | 34.0 | 45.5 | 26.5 |
| Direct SFT | 59.8 | 5.8 | 7.1 | 45.7 | 13.9 | 32.8 | 45.8 | 26.6 |
| *Other Open-source models* | | | | | | | | |
| DS-Distill-Qwen-1.5B | 83.1 | 29.9 | 21.7 | 66.3 | 15.9 | 35.8 | 35.5 | 31.7 |
| DeepSeek-R1-671B | 97.3 | 79.8 | 70.0 | 97.6 | 65.9 | 71.5 | 84.0 | 88.5 |

Vanilla mid-training on MegaMath-Web-Pro-Max and FineWeb-Edu for 40B tokens, as seen in Table 6. shows minimal impact on most benchmarks, with mathematical and general reasoning scores remaining largely stable. However, a notable decline in coding performance is observed. This suggests that given LLaMA-3.2-3B's extensive pre-training, additional vanilla training yields limited benefits and may even be detrimental to certain domains.

We also include DeepSeek's distillation models as reference, noting that these were trained on a substantially larger private dataset compared to the Mixture-of-Thoughts dataset, making direct comparisons less meaningful.

Table 7: Base model performance under constrained data setting across 5 datasets and their average. The total tokens in the raw documents are constrained to 10B.

| Model | # tokens | # epochs | GSM8k | MATH | BoolQ | MMLU | MMLU$_{pro}$ | Avg |
|-------|----------|----------|-------|------|-------|------|--------------|-----|
| Vanilla-8B | 40B | $\sim 4$ | 6.7 | 4.8 | 53.6 | 12.8 | 5.2 | 16.6 |
| TPT-8B | 40B | $\sim 1$ | **30.5** | **12.9** | **70.3** | **33.7** | **15.8** | **32.6** |

In Table 7, we present the final base model performance corresponding to Section 3.2 for the reader's reference.

---

[1]https://github.com/huggingface/lighteval

[2]https://github.com/mlfoundations/evalchemy

### A.5 LIMITATIONS AND FUTURE WORK

For future work, scaling to larger training corpora and model sizes represent a natural next step. Our current approach utilizes a single prompt template, but the integration of automatic prompt optimization techniques could further improve data quality. Additionally, exploring more powerful thinking generation models warrants further investigation.

### A.6 EXAMPLES

Table 8 to Table 10 present several examples of the generated thinking trajectories, which are generated using DeepSeek-R1-Distill-Qwen-7B as the thinking generation model.

### A.7 USE OF LARGE LANGUAGE MODELS

Large language models (LLMs) were used solely to assist with the writing of this paper. No LLMs were involved in generating the research ideas or analyzing the results.

918
919
920
921
922
923
924

Table 8: Generated thinking trajectory for the example shown in Figure 1 using DeepSeek-R1-Distill-Qwen-7B.

**Given Document:**
The largest positive integer $n$ for which $n^3 + 100$ is divisible by $n + 10$ is 890.

**Generated Thinking Trajectory (1054 tokens):**
Okay, so I need to figure out why the largest positive integer $n$ for which $n^3 + 100$ is divisible by $n + 10$ is 890. Hmm, let's start by understanding the problem.

First, the problem says that $n^3 + 100$ should be divisible by $n + 10$. In mathematical terms, that means when you divide $n^3 + 100$ by $n + 10$, there should be no remainder. Another way to think about this is using the Remainder Theorem. I remember that the Remainder Theorem states that if you divide a polynomial $f(n)$ by $(n - a)$, the remainder is $f(a)$. So, in this case, if I consider $f(n) = n^3 + 100$ and divide it by $(n + 10)$, which is the same as $(n - (-10))$, the remainder should be $f(-10)$.

Let me calculate that remainder. Plugging in $-10$ into $f(n)$:

$f(-10) = (-10)^3 + 100 = -1000 + 100 = -900$.
So the remainder is $-900$. For $n + 10$ to divide $n^3 + 100$ exactly, this remainder must be zero. But right now, it's $-900$, which isn't zero. That means $n + 10$ is a factor of $n^3 + 100 + 900$, which is $n^3 + 1000$.

Wait, let me check that again. If the remainder is $-900$, then $n^3 + 100 = (n + 10) * Q(n) - 900$, where $Q(n)$ is the quotient polynomial. So, for $n + 10$ to divide $n^3 + 100$, the remainder must be zero. Therefore, $(n + 10)$ must divide $-900$. Which means that $n + 10$ is a divisor of 900.

So, $n + 10$ must be a positive divisor of 900 because $n$ is a positive integer. Therefore, $n + 10$ can be any of the divisors of 900. To find the largest possible $n$, we need the largest divisor of 900 and then subtract 10.

What's the largest divisor of 900? Well, 900 itself is a divisor. So, $n + 10 = 900$ implies $n = 900 - 10 = 890$. That makes sense because the problem states that 890 is the largest such integer.

Wait, let me make sure I didn't skip any steps. So, starting from the Remainder Theorem, I found the remainder when dividing by $n + 10$, set it to zero, which led me to the conclusion that $n + 10$ must divide 900. Then, since we're looking for the largest $n$, we take the largest divisor, which is 900, subtract 10 to get $n = 890$.

Is there any possibility that I missed other divisors? For example, could there be a larger $n$ if I considered negative divisors? But $n$ is a positive integer, so $n + 10$ must also be positive. Therefore, negative divisors don't apply here. So, 900 is indeed the largest possible divisor in this context.

Just to double-check, let's plug $n = 890$ into the original expression and see if it's divisible by 900 (since $n + 10 = 900$).

Calculating $n^3 + 100$ when $n = 890$:
$890^3$ is a huge number, but since we're checking divisibility by 900, we can think in terms of modular arithmetic. Specifically, $890 \equiv -10 \pmod{900}$ because $890 = 900 - 10$.

So, $890^3 \equiv (-10)^3 = -1000 \pmod{900}$. Then, adding 100 gives $-1000 + 100 = -900$, which is congruent to $0 \pmod{900}$. Therefore, $n^3 + 100$ is divisible by 900, which confirms that $n = 890$ works.

Is there a possibility of a larger $n$? Well, the next divisor after 900 would be larger than 900, but 900 is the largest divisor of itself. So, there's no integer larger than 890 that satisfies the condition.

I think that covers it. I've considered the Remainder Theorem, found the necessary condition, identified the largest divisor, and confirmed that it works. I don't see any gaps in this reasoning.

925
926
927
928
929
930
931
932
933
934
935
936
937
938
939
940
941
942
943
944
945
946
947
948
949
950
951
952
953
954
955
956
957
958
959
960
961
962
963
964
965
966
967
968
969
970
971

Table 9: Example of generated thinking trajectory from MegaMath-Web-Pro-Max using DeepSeek-R1-Distill-Qwen-7B.

**Given Document:**

Complexity: This week, we explore two key classes of problems: P and NP. The learning objectives include:
- Defining the running time/time complexity of a DTM and NDTM
- Using big-O notation to express orders of growth
- Showing time bounds on multi-tape TMs and NDTMs simulated by DTMs
- Defining the classes P and NP
- Analyzing the cost of an algorithm using big-O notation
- Designing and analyzing verification for NP problems
- Explaining the P vs NP question

Pre-class Activity: Plot $x^2$ and $2^x$. Compare their growth rates by plotting $x^2/2^x$. If the values get smaller, $2^x$ grows faster than $x^2$.

Lecture Videos: PDF slides are available. This week, we focus on time complexity of decidable problems, concentrating on time and space required for computation.

Time Complexity: Not all problems are decidable. This week, we ask: "When a problem is decidable, how much does it cost to decide?" We prioritize time required and secondarily consider space needed.

... *(omitted for space reason)* ...

**Generated Thinking Trajectory (2325 tokens):**

Okay, so I'm trying to wrap my head around the concepts of P and NP, and how to analyze the time complexity of algorithms. Let me start by breaking down the information provided.

First, the context mentions that P and NP are classes of problems based on whether a deterministic or non-deterministic Turing machine (DTM or NTM) can solve them in polynomial time. Polynomial time means the running time is bounded by a polynomial function of the input size, like $O(n^2)$ or $O(n^3)$. So, if an algorithm's running time grows faster than any polynomial, it's not considered efficient for large inputs.

The pre-class activity suggests plotting $x^2$ and $2^x$, and then their ratio to see which grows faster. I remember that exponential functions like $2^x$ grow much faster than polynomial ones like $x^2$. So, $2^x$ will eventually dominate $x^2$ as $x$ increases, making the ratio $x^2/2^x$ approach zero. This tells us that exponential time is slower than polynomial time for large inputs.

... *(omitted for space reason)* ...

Wait, the Cook-Levin theorem states that SAT is NP-complete, meaning that any problem in NP can be reduced to SAT in polynomial time. So, if we can solve SAT in exponential time, then all NP problems can be solved in exponential time. But if $P = NP$, then SAT can be solved in polynomial time, implying all NP problems can be solved in polynomial time.

I should also consider how to verify NP problems. For example, in the traveling salesman problem (TSP), given a tour, you can verify if it's a valid tour and its cost in polynomial time. The decision version of TSP, which asks if there's a tour with cost less than a given value, is NP-complete. So, verifying a solution is easy, but finding the solution might not be.

In terms of practical implications, if $P = NP$, many problems that are currently considered intractable (like integer factorization, which is used in RSA encryption) could be solved efficiently. This would have major impacts on cryptography, logistics, optimization, etc. But most experts believe $P \neq NP$, meaning that some problems require exponentially long times to solve, even though their solutions can be verified quickly.

I also need to think about how to analyze algorithms. For each problem, I should determine the number of operations it performs as a function of the input size, then express this using big-O notation. For example, matrix multiplication is $O(n^3)$ for the standard algorithm, but more efficient algorithms like Strassen's algorithm are $O(n^{2.807})$, which is better.

In summary, the key points are:
- P: Problems solvable by DTM in polynomial time.
- NP: Problems solvable by NTM in polynomial time, or problems where a solution can be verified by DTM in polynomial time.
- Big-O notation helps in analyzing and comparing the efficiency of algorithms.
- Simulating NTMs with DTMs introduces a polynomial overhead.
- The P vs NP question remains unresolved but has significant implications for computational efficiency and many real-world applications.

... *(omitted for space reason)* ...

Table 10: Example of generated thinking trajectory from FineWeb-Edu using DeepSeek-R1-Distill-Qwen-7B.

**Given Document:**

December 16, 2022:

Scientists have rediscovered the black-naped pheasant pigeon, an extremely rare bird, in Papua New Guinea. The bird was last seen around 140 years ago and was believed to be extinct. This year, The Search for Lost Birds team, a collaboration between Bird-Life International, American Bird Conservancy, and Re:wild, conducted an intense search for the bird for a month on Fergusson Island in Papua New Guinea.

The team set multiple camera traps on Mount Kilkerran and interviewed hunters who had spotted the pigeon and heard its call. Footage of the bird was captured in the last days before the researchers planned to leave the island. Did you know?

The black-naped pheasant pigeon is critically endangered, and scientists hope that its rediscovery will prevent the species from becoming extinct.

**Generated Thinking Trajectory (1059 tokens):**

Okay, so I'm trying to figure out the key points from this article about the rediscovery of the black-naped pheasant pigeon. Let's start by reading through it again to make sure I understand the main details.

The article says that scientists found this rare bird in Papua New Guinea for the first time in over a century. It was last seen around 140 years ago and was thought to be extinct. That's really significant because it shows how some species can survive long periods without being noticed.

The team involved is The Search for Lost Birds, which is a collaboration between Bird-Life International, American Bird Conservancy, and Re:wild. They spent a month searching on Fergusson Island. I'm not sure where exactly Fergusson Island is, but it's probably a significant habitat for this bird.

They used camera traps on Mount Kilkerran and interviewed hunters who saw the pigeon and heard its call. That makes me think that they were able to gather some direct observations and maybe even talk to people who might have seen it before, which is a great way to gather information.

The bird was last seen around 140 years ago, so this isn't just a recent discovery. It shows that even in supposedly stable environments, species can disappear. The rediscovery is hopeful because the pigeon was believed extinct, so this might prevent its extinction.

There's also a factoid about the pigeon being critically endangered. That term means it's at high risk of extinction, so the rediscovery is a big deal because it might help prevent that. The article mentions that scientists hope this will prevent extinction, which is a positive outcome.

Now, thinking about the deeper implications, why is this pigeon so important? Maybe it's a keystone species or plays a significant role in the ecosystem. If it's extinct, it could have cascading effects on other species that depend on it. Rediscovering it might help in conservation efforts and restoring its population.

... *(omitted for space reason)* ...

In terms of conservation efforts, this discovery could lead to new research on the pigeon's behavior, habitat preferences, and threats. Understanding its ecology better can inform more effective management strategies.

I'm also considering the broader conservation implications. If this pigeon is one of several critically endangered species rediscovered, it might indicate a positive trend in some regions, though it's still a cause for concern globally.

The fact that it's a pheasant pigeon might mean it's a flightless bird, which could make it more vulnerable to predators or environmental changes that affect non-flying species.

I'm curious about the name "black-naped pheasant pigeon." I know pheasants are usually flightless, but pigeons are flightless too. So maybe this species has a unique combination of traits that make it more adaptable or less visible to predators.

The team's work involved collaboration with Re:wild, which I believe focuses on citizen science and community conservation. So maybe they engaged local communities in the search, which is a good practice for ensuring sustainable conservation efforts.

In summary, the key points are the rediscovery of a critically endangered species after a long period of extinction, the methods used to find it, the significance of the species to conservation efforts, and the potential implications for the ecosystem and local communities. The article highlights the importance of international cooperation and community involvement in wildlife conservation.