# OpenReview forum: "Thinking Augmented Pre-training"
_ICLR.cc/2026/Conference — Submitted to ICLR 2026_

### Official Review · Reviewer_cAFT · 2025-10-29

**Soundness:** 3
**Presentation:** 3
**Contribution:** 2
**Rating:** 2
**Confidence:** 3

**Summary:**

This paper introduces a simple approach to improve the data efficiency of LLMs, namely Thinking-augmented Pre-Training (TPT). It augments raw text corpora with automatically generated *thinking trajectories*, which involves step-by-step reasoning rationales produced by LLMs. This augmentation effectively increases the volume of the training data, and makes high-quality tokens more learnable with the step-by-step reasoning trajectories. The paper applies TPT across diverse training regimes (from-scratch pre-training, data-constrained settings, and mid-training from strong open-source checkpoints) using up to 100B tokens. Experiments show gains across model sizes and benchmarks, with reported improvements of >10% on challenging reasoning tasks.

**Strengths:**

1. It is valuable to improving data efficiency in an era of diminishing returns from scaling data. The paper directly acquires training data with explicit reasoning decomposition.
2. The method is simple, and does not require human annotation. It can be easy to obtain the data and useful in multiple training phases.
3. The paper reports consistent result improvements, which demonstrates the effectiveness of the proposed method.

**Weaknesses:**

1. (Major) The novelty is somewhat limited. The core idea to train on self-generated rationales has precedents in several researches, such as distillation, synthetic data, and self-explanation. The paper ought to more clearly differentiate TPT from these approaches, especially in terms of training stage, data construction, and scalability.
2. (Major) The quality and reliability of generated trajectories are not guaranteed. The method relies entirely on LLMs to generate reasoning traces. However, there is no analysis of error propagation (e.g., how often are the trajectories incorrect or unhelpful?), nor any filtering or verification mechanism. This raises concerns about potential data pollution.
3. (Minor) It is not clear about the generalization beyond reasoning tasks, such as language modeling perplexity, commonsense QA, or stylistic generation. It would be valuable to assess whether TPT helps (or hurts) performance on non-reasoning tasks.

**Questions:**

Please see strengths and weaknesses.

Besides, how to ensure the quality of the automatically generated thinking trajectories? Did you perform human or automatic evaluation (e.g., faithfulness, coherence) on these trajectories? Besides, the teacher model capability may also affect TPT performance.

---

> ### Author Response · Authors · 2025-11-24
> **Author Response**
>
> Thank you for your valuable feedback!
>
> >"The novelty is somewhat limited. The core idea to train on self-generated rationales has precedents in several researches, such as distillation, synthetic data, and self-explanation. The paper ought to more clearly differentiate TPT from these approaches, especially in terms of training stage, data construction, and scalability."
>
> There are several existing papers that include generated rationales for model training, as we discussed in the Related Work section. Our proposed method has several key distinctions:
> - **Training Scale and Configurations**: We substantially scale up to 100B training tokens (vs. 150M-8B in prior work [2,3]) and validate across multiple training configurations (pre-training, mid-training, SFT) with broader benchmarks spanning mathematics, coding, and general reasoning.
> - **Simplicity**: TPT is a considerably simpler approach requiring neither meticulous prompt engineering [2] nor complex iterative training algorithms like EM [3].
> - **Computational Efficiency**: Our method is more efficient than RPT [1], which requires substantially more compute due to online rollouts and token-level reinforcement learning paradigm.
>
> References:
>
> [1] Dong, Qingxiu, et al. "Reinforcement Pre-Training." arXiv preprint arXiv:2506.08007 (2025).
>
> [2] Ishibashi, Yoichi, Taro Yano, and Masafumi Oyamada. "Mining hidden thoughts from texts: Evaluating continual pretraining with synthetic data for llm reasoning." arXiv preprint arXiv:2505.10182 (2025).
>
> [3] Ruan, Yangjun, et al. "Reasoning to learn from latent thoughts." arXiv preprint arXiv:2503.18866 (2025).
>
> >"how to ensure the quality of the automatically generated thinking trajectories? Did you perform human or automatic evaluation (e.g., faithfulness, coherence) on these trajectories?"
>
> We appreciate the concern regarding the quality of the automatically generated thinking trajectories. We employed several strategies to ensure and assess their quality:
>
> * **Indirect Evaluation via Final Performance**: The most critical validation of the trajectory quality is its impact on the final model performance. The consistent and substantial gains achieved by TPT strongly suggest that the generated trajectories are, on average, highly helpful to acquire enhanced reasoning skills.
> * **Qualitative Inspection**: For a more direct inspection, we manually checked a small sample of the generated trajectories and found the vast majority to be coherent and informative. Several representative examples are provided in Appendix A.6 of the paper, which we hope will offer readers a clearer sense of the generated data quality.
>
> >"It would be valuable to assess whether TPT helps (or hurts) performance on non-reasoning tasks."
>
> Thanks for your suggestion, we did evaluate TPT on non-reasoning tasks for the pre-training from scratch setting in Table 1, including the BoolQ and MMLU benchmarks. To further address your concern, we evaluated on three additional tasks: Winogrande, CommonsenseQA, and PIQA. The full results are as follows:
>
> | Model                    | MATH | BoolQ | MMLU | Winogrande | CommonsenseQA | PIQA |
> |--------------------------|------|-------|------|------------|---------------|------|
> | Vanilla-8B (100B tokens) | 9.1  | 66.5  | 26.2 | 53.9       | 44.6          | 73.2 |
> | TPT-8B (100B tokens)     | 21.8 | 75.0  | 46.7 | 54.5       | 43.0          | 72.4 |
>
> Despite TPT-8B seeing less than half raw tokens compared to Vanilla-8B due to the augmented thinking trajectories, the results show:
>
> * **Substantial Gains on Reasoning**: TPT yields significant improvements on the main reasoning task (**MATH**).
> * **Non-Reasoning Tasks Maintained**: Performance on tasks targeting commonsense, knowledge memorization, and language understanding (Winogrande, CommonsenseQA, PIQA, BoolQ, MMLU) is maintained.
>
> This suggests that TPT **primarily enhances reasoning skills without detracting from the model's general language and knowledge capabilities**.

---

### Official Review · Reviewer_6v2k · 2025-10-31

**Soundness:** 2
**Presentation:** 3
**Contribution:** 3
**Rating:** 2
**Confidence:** 4

**Summary:**

This paper introduces thinking augmented pretraining, a pretraining method which adds thinking tokens generated by a strong model to the original pretraining data. The motivation is that by using these traces, the model will learn how to think during the pretraining phase. The authors demonstrate that TPT improves performance on reasoning tasks and can increase data efficiency 3x shown on both pre-training and mid-training.

**Strengths:**

- Nice idea for "re-writing" the pretraining data to include thinking tokens
- Method is simple, well explained, and can be easily implemented
- Figure 2 (right) clearly shows that this method is better than vanilla pretraining
- The abundant data results are nice.

**Weaknesses:**

- My biggest concern is that it doesn't show it help for post-training. Since it's used to generate the thinking tokens, we can assume that we already have a post-trained model. It's not shown that this helps over that original model (i know that it's a bigger model, but it still needs to be shown that this method helps post-training in general).
- The SFT dataset is R1 distilled, and the TPT thinking traces are distilled-R1). So it seems like this is primarily just R1 thinking distillation. This is ok in general because it shows that distilling into pretraining helps, but I think more experiments need to be conducted to show that this full pipeline with postraining works.
- "Midtraining" experiments are done on models that are already post-trained (llama-instruct and qwen-instruct). Ideally this should be done on base models
- The analysis and ablations don't really give insight to why this method is performing better than pretraining on the raw MegaMath tokens. It seems like a lot has to do with pure thinking distillation. I wonder if you can elicit this behavior with pure synthetic data from DS-Distill-Qwen-7B and not using MegaMath at all. Or re-writing the data with DS-Distill-Qwen-7B, instead of "appending".

**Questions:**

- Would lab that's launching a large-scale pretraining run actually do this? The experiments are done on small scale models, which is fine, but it's unclear what the benefits would be as you scale up
- Is "constrained data" really a problem for pretraining? This is mostly done for large-scale runs\
- The models in Table 2 are unclear to me. Do you pretrain the baselines with the same pretraining data that you use for TPT (MegaMath-Web-Pro-Max), but without the thinking tokens? If not, I don't think this is a fair baseline.
- How important is the thinking generator model (DS-Distill-Qwen-7B)? Can you easily swap it for other models?

---

> ### Author Response · Authors · 2025-11-24
> **Author Response**
>
> Thank you for your thoughtful review and constructive feedback!
>
> >"My biggest concern is that it doesn't show it help for post-training. Since it's used to generate the thinking tokens, we can assume that we already have a post-trained model. It's not shown that this helps over that original model"
>
> We respectfully disagree with the assertion that TPT is bounded by the original thinking generation model. Our **mid-training results in Table 2** demonstrate that TPT-Qwen2.5-7B (our student model) can **outperform** the thinking generator model, DS-Distill-Qwen-7B, on most benchmarks after SFT.
>
> This demonstrates that TPT is not merely a form of direct distillation. The student model still learns from the human data distribution *enhanced* by the teacher's thinking process, allowing it to potentially extrapolate beyond the teacher's capabilities.
>
> This framework naturally suggests a **model-data co-evolution approach**, where an improved TPT model could be used iteratively to generate even better thinking trajectories, continuously bootstrapping the system.
>
> >""Midtraining" experiments are done on models that are already post-trained (llama-instruct and qwen-instruct). Ideally this should be done on base models"
>
> Thank you for pointing out the potential ambiguity. As detailed in **Appendix A.1**, the models used in our mid-training experiments are actually **initialized from the base models**. We have revised the main text to make this clearer.
>
> >"Would lab that's launching a large-scale pretraining run actually do this?"
>
> We do not have information on this, but we believe it is worth a try given the promising results we observed.
>
> >"Is "constrained data" really a problem for pretraining?"
>
> While the total volume of general web text is massive, high-quality data is inherently limited and expensive to obtain.
>
> The 10B unique token limit in our paper is an academic setting designed to **simulate this constrained data scenario**. Our method provides a simple, effective solution to improve data efficiency and help mitigate the constrained data problem.
>
> >"Do you pretrain the baselines with the same pretraining data that you use for TPT (MegaMath-Web-Pro-Max), but without the thinking tokens? If not, I don't think this is a fair baseline."
>
> We include such a baseline in Table 6 from Appendix A.4. The "Vanilla mid-training -> SFT" row first mid-trains on the dataset without thinking tokens. We did not observe any gain compared to directly performing SFT. The likely reason is that popular models already include a similar mid-training stage. Therefore, we did not include this baseline in Table 2 to save space.
>
> >"How important is the thinking generator model (DS-Distill-Qwen-7B)? Can you easily swap it for other models?"
>
> Our ablation study in Table 3 directly addresses this. We demonstrated that TPT is **not overly sensitive** to the choice of the thinking generator model for the tested model sizes. We can easily swap to a smaller 1.5B thinking generator model, and still observe significant gains from TPT. Using a smaller thinking generator in fact leads to slightly better results, indicating that a smaller student-teacher gap can be beneficial for effective distillation.

---

### Official Review · Reviewer_7Wjm · 2025-10-31

**Soundness:** 2
**Presentation:** 3
**Contribution:** 2
**Rating:** 4
**Confidence:** 2

**Summary:**

The paper introduces Thinking Augmented Pre-Training (TPT), a data engineering method designed to improve the data efficiency and reasoning capabilities of Large Language Models (LLMs). The method involves using an "off-the-shelf" teacher LLM to generate a "thinking trajectory" (e.g., a step-by-step rationale) for each document in an existing pre-training corpus. This trajectory is then concatenated with the original document to create an augmented dataset. A new LLM is then trained from scratch (or mid-trained) on this augmented data using a standard next-token prediction objective. The authors claim this "simple and scalable" method improves data efficiency by a factor of 3  and enables an 8B model to achieve reasoning performance superior to strong baselines (like LLaMA-3.1-8B-Instruct) trained on 150x more data.

**Strengths:**

1. The paper is well-written, and the method is described in an easy-to-understand manner.

2. The core idea of TPT is conceptually simple and elegant.

3. The experimental results are significant.

**Weaknesses:**

1. The "abundant data" pre-training experiment, training an 8B parameter model from scratch on only 100B tokens , is severely undertrained by modern standards (where models are trained on 10T+ tokens ). This limited training budget makes it difficult to assess the true, long-term value of the TPT method. The "3x data efficiency"  claim may be an artifact of this specific, undertrained regime, where the model converges faster on the "thinking" data, rather than an indicator of a higher asymptotic performance.

2. The mid-training experiments in Table 2  lack the most crucial control group. The paper compares its TPT mid-training models (which receive 100B tokens of TPT data plus SFT ) against base models that only receive SFT (e.g., "OpenR1-LLAMA-3B*"). This comparison is confounded. It is impossible to know if the observed gains come from the nature of TPT data or simply from the act of continual pre-training on 100B additional tokens. A necessary baseline—a model mid-trained on 100B tokens of the original, non-augmented dataset before SFT—is not included in this table, rendering the conclusions about TPT's mid-training efficacy unsubstantiated.

3. The paper's key results table (Table 2)  contains highly anomalous scores that undermine the evaluation's credibility.
    - The TPT-LLAMA-3B model scores a mere 0.3 on GSM8K, which is nonsensical for a model that simultaneously scores 75.5 on MATH-500 and 18.6 on AIME24. This score is lower than its non-thinking OpenR1-LLAMA-3B* counterpart (3.8) and suggests a catastrophic failure in either the SFT process or the evaluation pipeline.
    - For stronger models, the scores are inconsistent. The DS-Distill-Qwen-7B reference model achieves a powerful 53.2 on AIME24 but only 90.7 on GSM8K. Models with such strong Olympiad-level math reasoning would be expected to be at or near saturation on a grade-school math benchmark like GSM8K. This disconnect suggests the SFT hyperparameters were not properly tuned, leading to unreliable results.

4. The paper's premise is that TPT teaches models to reason. However, the method is functionally a simple distillation, where a student model is trained to reproduce the text generated by a teacher. It is a common finding that augmenting pre-training with such data improves performance on pre-train evaluations (i.e., the model gets good at mimicking the "thinking" style) but fails to generalize or show robust gains after post-training (SFT). The paper does not provide sufficient evidence (e.g., post-train evaluations on held-out reasoning tasks) to disprove this alternative explanation.

**Questions:**

1. Can the authors justify the 100B token budget for an 8B model? Given this is far from convergence, how can the authors be confident that the TPT model isn't just learning faster on structured data, rather than learning better or reaching a higher final performance?

2. Can the authors please explain the 0.3 GSM8K score for the TPT-LLAMA-3B model?

3. Given the inconsistent results, what steps were taken to ensure optimal SFT hyperparameter tuning for all models being compared, including the baselines? The low GSM8K scores suggest the SFT process may have been flawed, potentially biasing the comparison.

---

> ### Author Response · Authors · 2025-11-24
> **Author Response**
>
> Thank you for your detailed review and constructive feedback!
>
> >"The paper's key results table (Table 2) contains highly anomalous scores that undermine the evaluation's credibility. The TPT-LLAMA-3B model scores a mere 0.3 on GSM8K, which is nonsensical. For stronger models, the scores are inconsistent. The DS-Distill-Qwen-7B reference model achieves a powerful 53.2 on AIME24 but only 90.7 on GSM8K."
>
> It seems there might be a slight **misreading of the results in Table 2**. The TPT-LLaMA-3B model actually achieves an accuracy of **81.6** on GSM8K, not 0.3. **We kindly ask you to double-check the values in the table.**
>
> The DS-Distill-Qwen-7B model's 90.7 on GSM8k is in fact normal and is consistent with results reported in prior works. For example, the MobileLLM-R1 paper [1] reports similar GSM8k scores for models that achieve around 50 on AIME24 in Table 7. While we agree that 7B models with strong reasoning should nearly saturate GSM8K, the benchmark remains valuable, especially for evaluating smaller models, despite known issues like occasional label errors and ambiguous problem descriptions.
>
> References:
>
> [1] Zhao, Changsheng, et al. "MobileLLM-R1: Exploring the Limits of Sub-Billion Language Model Reasoners with Open Training Recipes." arXiv preprint arXiv:2509.24945 (2025).
>
> >"The "abundant data" pre-training experiment, training an 8B parameter model from scratch on only 100B tokens , is severely undertrained by modern standards (where models are trained on 10T+ tokens ). Can the authors justify the 100B token budget for an 8B model? Given this is far from convergence, how can the authors be confident that the TPT model isn't just learning faster on structured data, rather than learning better or reaching a higher final performance?"
>
> We completely agree that **100B tokens is insufficient for full convergence** of an 8B model. Training on 10T+ tokens requires enormous computational resources that are unfortunately beyond our current capabilities. Figure 1 shows no sign of saturation up to the 100B training tokens, indicating **potential for further scaling**.
>
> Critically, our mid-training experiments (Table 2) add a negligible amount of data ($\approx$ 100B tokens) when compared to the original models' massive pre-training budgets (e.g., Qwen-2.5 at 18T tokens). The consistent and substantial gains observed from TPT in this regime strongly suggest that **TPT is helping the model learn better reasoning capabilities**, rather than merely facilitating a faster convergence on structured data.
>
> >"The mid-training experiments in Table 2 lack the most crucial control group. A necessary baseline—a model mid-trained on 100B tokens of the original, non-augmented dataset before SFT—is not included in this table, rendering the conclusions about TPT's mid-training efficacy unsubstantiated."
>
> We did, in fact, include this necessary control group. The results are presented in **Table 6 of Appendix A.4**. The "Vanilla mid-training -> SFT" row represents the model mid-trained on the original dataset before SFT. We observed **no noticeable gain** from this control experiment compared to directly performing SFT. The likely reason is that the foundational base models already incorporate a similar pre-training or mid-training stage. To conserve space in the main paper, we opted not to include this non-impactful baseline in Table 2.
>
> >"It is a common finding that augmenting pre-training with such data improves performance on pre-train evaluations (i.e., the model gets good at mimicking the "thinking" style) but fails to generalize or show robust gains after post-training (SFT). The paper does not provide sufficient evidence (e.g., post-train evaluations on held-out reasoning tasks) to disprove this alternative explanation."
>
> We must emphasize that our **main mid-training results in Table 2 are all post-SFT evaluations**, and they show consistent gains *after* the SFT phase, which directly contradicts the alternative explanation you posit.
>
> Furthermore, to demonstrate generalization to held-out reasoning tasks, we can point to the fact that we **did not include code data in our TPT mid-training**, yet the evaluations still show meaningful **gains on coding benchmarks** such as HumanEval and LiveCodeBench. This transfer of capability from TPT data to a distinct domain is strong evidence that **TPT helps the model acquire better, generalized reasoning skills** rather than just mimicking the "thinking" style.

---

> > ### Comment · Reviewer_7Wjm · 2025-11-28
> >
> > Thank you very much for the detailed response and clarifications. I apologize for the oversight regarding the GSM8K score.
> >
> > I agree that the appendix baseline with vanilla mid-training helps clarify that the gains are not simply from adding more tokens. However, I still see a remaining gap relative to current practice.
> >
> > Many recent mid-training recipes for reasoning models already include explicit reasoning-trace data and are tuned to make mid-training itself clearly beneficial, especially when followed by a long CoT SFT stage. For example, SmolLM3 [1] performs a dedicated “reasoning mid-training” stage on 35B tokens from OpenThoughts3-1.2M and a subset of NVIDIA’s Llama-Nemotron post-training dataset with R1 reasoning traces, before SFT and alignment.
> > Similarly, OctoThinker [2] systematically studies mid-training mixtures that combine high-quality math web corpora with both short-CoT and long-CoT QA data such as OpenR1-Math-220K.
> >
> > In this context, the “vanilla mid-training” control in Appendix Table 6, which yields almost no improvement, seems under-tuned compared to a more realistic baseline where mid-training is strong enough to be useful on its own.
> > I would encourage the authors to add such a stronger baseline (e.g., curated CoT-based mid-training without TPT-style document-level trajectories).
> >
> > Given that my primary concerns regarding the validity of the results and the baselines have been addressed, I am considering updating my score.
> >
> > [1] Bakouch, Elie, et al. "SmolLM3: smol, multilingual, long-context reasoner." Hugging Face Blog (2025). https://huggingface.co/blog/smollm3
> >
> > [2] Wang, Zengzhi, et al. "Octothinker: Mid-training incentivizes reinforcement learning scaling." arXiv:2506.20512.

---

> > > ### Author Response · Authors · 2025-12-03
> > >
> > > Thank you for the constructive follow-up comments and the suggestions regarding stronger mid-training baselines like SmolLM3 and OctoThinker. We fully agree that a more carefully curated mid-training stage can yield better results than our vanilla control.
> > >
> > > We believe that comparing TPT against these existing, high-performing methods, while valuable for future work, requires acknowledging a few key differences in scope and approach:
> > >
> > > * **Scalability and Data Type:** The core distinction of our TPT method is its inherent scalability. It operates directly on **massive raw document collections** without the need for the labor-intensive **question-style prompt curation** that limits the scale of SFT-derived datasets used in the reasoning mid-training stage of models like SmolLM3.
> > > * **Post-Training Focus:** Our work primarily investigates the impact of TPT when followed by a standard and stable SFT stage. In contrast, OctoThinker specifically focuses on how mid-training affects a subsequent, often less stable, RL stage (e.g., in the R1-zero setting). We already incorporate high-quality math data, such as a derivative of OctoThinker's MegaMath-Web-Pro-Max, into our primary mid-training mix.
> > >
> > > Thank you again for your insightful feedback!

---

### Official Review · Reviewer_Kpxd · 2025-11-01

**Soundness:** 3
**Presentation:** 3
**Contribution:** 2
**Rating:** 6
**Confidence:** 3

**Summary:**

This work proposes a novel way to approach pretraining by augmenting existing data with LLM generated thoughts. Authors employ a strong, pre-trained reasoning model to add more data into existing corpus, and execute pretraining with that data. They show how such pretrained model is much better than counter parts trained on similarly sized data without augmented data inside, as well as better than models trained on much more human data.
Further, they show how fine-tuning with such data brings substantial improvements on the benchmarks related to reasoning and other LLM capabilities related to solving some tasks. Finally, they performed ablations with different model families and sizes, as well as different ways to generate trajectories.

**Strengths:**

Recently there have been a massive exodus of papers in this direction. All of them somehow generate more high quality data from a strong reasoning model to add into an existing pretraining corpus. Different publications tend to differentiate in a ways of adding this data, types of prompts they use, and how the resulting augmented data is used for pre or mid training.

The main strength of this work is the experimental results and ablations showing that their model is better than pretrained models that did not see such data. There been too many simultaneous work in this direction since it was an obvious move, but I believe that all of them did it with originality in mind.

**Weaknesses:**

My main concern here is that we aim at improving the benchmark scores where the model we use to generate more data from already excel at those benchmarks. I might be wrong here, but this might be creating a vicious loop between the distributions of data we fit and "seemingly unknown" evaluation sets we test it on.
We may always find a strong teacher to augment such datasets and then show improvements on training much smaller models in modest data regimes. Then how would we improve the strongest teachers? To answer that questions we might want to experiment in the setup where the teacher is as strong as the mid training checkpoint we start from. Authors performed some ablation experiments in this direction (using smaller model for trajectory generation), but that smaller model is very mighty as its distilled from R1 model.

**Questions:**

1. Do you think this data augmentation is a prominent approach to go with a frontier model development?

---

> ### Author Response · Authors · 2025-11-24
> **Author Response**
>
> Thank you for your valuable feedback!
>
> >"We may always find a strong teacher to augment such datasets and then show improvements on training much smaller models in modest data regimes. Then how would we improve the strongest teachers? To answer that questions we might want to experiment in the setup where the teacher is as strong as the mid training checkpoint we start from."
>
> We agree that this is an important question—whether this approach can help improve the strongest teachers.
>
> Crucially, the success of TPT is **not strictly bounded by the teacher model's performance**. In our main mid-training experiments (Table 2), we use DS-Distill-Qwen-7B as the teacher model to generate thinking trajectories. The resultant student model, TPT-Qwen2.5-7B, in fact **outperforms this teacher model on most benchmarks** after SFT.
>
> This demonstrates that TPT is not merely a form of direct distillation. The student model still learns from the human data distribution *enhanced* by the teacher's thinking process, allowing it to potentially extrapolate beyond the teacher's capabilities. This suggests a mechanism for improvement even when the teacher is highly capable.
>
> >"My main concern here is that we aim at improving the benchmark scores where the model we use to generate more data from already excel at those benchmarks."
>
> In fact, there is still **significant room for improvement for our tested model sizes** on these benchmarks, even for models considered "strong."
>
> For example, our best model, TPT-Qwen2.5-7B, achieves a score of 57.5 on AIME24, while many frontier models (e.g., in the GPT-5 class) can achieve scores over 90 on the same benchmark.
>
> On the other hand, even if we choose a frontier model as a teacher for trajectory generation, such efforts are still meaningful. Distilling knowledge into **smaller student models** results in models that are **significantly more efficient and practical** for widespread real-world deployment.
>
> >"Do you think this data augmentation is a prominent approach to go with a frontier model development?"
>
> **Yes**, we believe that incorporating TPT is definitely worth a try and holds significant promise for frontier model development. It provides a structured, scalable way to introduce high-quality reasoning and problem-solving processes.

---

### Meta-Review · Area_Chair_k2nS · 2026-01-11

**Summary:**

TPT proposes augmenting pretraining data with thinking trajectories generated by a strong reasoning model (DS-Distill-Qwen-7B), claiming 3x data efficiency improvement. While the method is simple and the paper is well-written, significant concerns remain about novelty and experimental rigor. The core idea of training on self-generated rationales has clear precedents (distillation, synthetic data, Phi-series models), and the paper does not sufficiently differentiate TPT from these approaches. The method appears to be primarily R1-style distillation applied to pretraining. Key experimental issues include: (1) 100B tokens for an 8B model is undertrained, making it unclear whether TPT learns better or just converges faster on structured data; (2) the vanilla mid-training baseline (Appendix Table 6) showing no gain suggests it was under-tuned compared to modern mid-training recipes (SmolLM3, OctoThinker); (3) loss curves between TPT and vanilla pretraining are not directly comparable since they train on different data distributions. The quality and reliability of generated trajectories lack verification mechanisms.

**Reviewer Concerns:**

Partially addressed: (1) Student outperforming teacher (TPT-Qwen2.5-7B > DS-Distill-Qwen-7B) suggests TPT is not merely bounded distillation; (2) GSM8K score confusion resolved (7Wjm misread table); (3) Vanilla mid-training baseline exists in Appendix Table 6; (4) Non-reasoning task evaluations added showing capabilities maintained. Outstanding: (1) Limited novelty remains the central concern—two reviewers gave score of 2 citing precedents in distillation/synthetic data; (2) 7Wjm noted the vanilla mid-training baseline is under-tuned compared to modern recipes (SmolLM3, OctoThinker use curated CoT data); (3) No verification/filtering mechanism for trajectory quality; (4) Scalability to frontier models and whether this would actually be adopted remains speculative; (5) The two lowest-scoring reviewers (6v2k, cAFT) did not yet respond to rebuttals.

**Reviewer Scores:**

Kpxd (6): concern about teacher-student loop addressed, score likely unchanged. 7Wjm (4): Acknowledged concerns addressed and considered updating score, but raised new concern about weak baseline compared to SmolLM3/OctoThinker; score potentially 4 or 6. 6v2k (2): core concerns about R1 distillation and post-training value not fully resolved, score likely unchanged. cAFT (2): novelty and trajectory quality concerns partially addressed but fundamental novelty concern remains, score likely unchanged or at most 4.

---

### Decision · Program_Chairs · 2026-01-26

Reject